# Iron Starvation Induces Ferricrocin Production and the Reductive Iron Acquisition System in the Chromoblastomycosis Agent *Cladophialophora carrionii*

**DOI:** 10.3390/jof9070727

**Published:** 2023-07-05

**Authors:** Alexandre Melo Bailão, Kassyo Lobato Potenciano da Silva, Dayane Moraes, Beatrix Lechner, Herbert Lindner, Hubertus Haas, Célia Maria Almeida Soares, Mirelle Garcia Silva-Bailão

**Affiliations:** 1Instituto de Ciências Biológicas, Universidade Federal de Goiás, Goiânia 74690-900, Brazil; kassyo.potenciano@hotmail.com (K.L.P.d.S.); dayane123@ufg.br (D.M.); celia@ufg.br (C.M.A.S.); mgsbailao@ufg.br (M.G.S.-B.); 2Institute of Molecular Biology/Biocenter, Medical University of Innsbruck, 795J+RF Innsbruck, Austria; beatrix.lechner@i-med.ac.at (B.L.); hubertus.haas@i-med.ac.at (H.H.); 3Institute of Medical Biochemistry/Biocenter, Medical University of Innsbruck, 795J+RF Innsbruck, Austria; herbert.lindner@i-med.ac.at

**Keywords:** siderophores, hydroxamate, nutritional immunity, black fungi

## Abstract

Iron is a micronutrient required by almost all living organisms. Despite being essential, the availability of this metal is low in aerobic environments. Additionally, mammalian hosts evolved strategies to restrict iron from invading microorganisms. In this scenario, the survival of pathogenic fungi depends on high-affinity iron uptake mechanisms. Here, we show that the production of siderophores and the reductive iron acquisition system (RIA) are employed by *Cladophialophora carrionii* under iron restriction. This black fungus is one of the causative agents of chromoblastomycosis, a neglected subcutaneous tropical disease. Siderophore biosynthesis genes are arranged in clusters and, interestingly, two RIA systems are present in the genome. Orthologs of putative siderophore transporters were identified as well. Iron starvation regulates the expression of genes related to both siderophore production and RIA systems, as well as of two transcription factors that regulate iron homeostasis in fungi. A chrome azurol S assay demonstrated the secretion of hydroxamate-type siderophores, which were further identified via RP-HPLC and mass spectrometry as ferricrocin. An analysis of cell extracts also revealed ferricrocin as an intracellular siderophore. The presence of active high-affinity iron acquisition systems may surely contribute to fungal survival during infection.

## 1. Introduction

Iron is necessary for essentially all living organisms. Due to its oxidation states, Fe^2+^ and Fe^3+^, this metal has the ability to gain or lose electrons, an important feature in many cellular processes, including cellular respiration [1]. Regardless of its essentiality, the Fe^3+^ resulting from the oxidation of Fe^2+^ is insoluble in water and with a neutral pH [2,3]. In addition, iron in excess triggers, via a Fenton/Haber Weiss reaction [4], the formation of hydroxyl radicals that are highly reactive and cause cell damage. Therefore, iron acquisition and homeostasis are based on ferric iron solubilization, transport across the plasma membrane into the cytosol and regulation of the uptake and utilization processes in order to avoid toxicity.

Pathogenic fungi face an iron-deprived scenario inside their hosts, which employ diverse strategies of nutritional immunity to decrease plasma and phagosome iron contents [5,6]. The relevance of this metal in the host–pathogen relationship is demonstrated by the increase in the frequency and severity of infections in people with iron overload [7]. To counteract the low availability, fungi have evolved high-affinity strategies of iron acquisition that can be used in parallel. These include the siderophore-mediated iron uptake and the reductive iron acquisition system (RIA).

Siderophores are low-molecular-weight iron chelators that bind Fe^3+^ with high affinity, solubilizing this ion [8]. These molecules are classified into three groups: carboxylates, catecholates and hydroxamates. The last includes the type of siderophores produced by fungi. Hydroxamate biosynthesis is well defined in *Aspergillus fumigatus* and is conserved in many fungal species [9]. In the first step, the ornithine oxygenase SidA catalyzes the hydroxylation of ornithine, a nonproteinogenic amino acid. Hydroxyornithine is then acylated to form the hydroxamate group. This step is catalyzed either by SidF or SidL, depending on the source of the acyl group, and the pathway to produce different siderophores is split. SidF catalyzes the incorporation of anhydromevalonyl-CoA to the hydroxyornithine, generating fusarinines and coprogens, while SidL defines the formation of ferrichromes siderophores (rhodotorulic acid, ferrichrome and ferricrocin) by the addition of an acetyl-CoA. Next, the non-ribosomal peptide synthetases (NRPSs; SidC and SidD) conjugate the hydroxamate groups via ester (fusarinine C) or peptide bonds (ferrichrome-type siderophores, such as ferricrocin and ferrichrome). In *A. fumigatus*, triacetylfusarinine C (TAFC) is synthesized from fusarinine C through the action of SidG, an acetyl transferase [10]. Interestingly, siderophore biosynthesis is connected to the ergosterol pathway via the action of the acyl-CoA ligase SidI and enoyl-CoA hydratase SidH [11]. The internalization of the siderophore-Fe^3+^ complex is usually mediated by transporters of the SIT subfamily (siderophore-iron transporter, subfamily 16), which belongs to the Major Facilitator Superfamily (MFS) [12].

As Fe^3+^ is insoluble at physiological pH in the presence of oxygen, its reduction to Fe^2+^ is necessary so that iron becomes soluble and is more easily captured. This RIA system is well characterized in *Saccharomyces cerevisiae* and involves two sequential steps. Fe^3+^ is initially reduced through the action of ferric reductase, FRE1 and FRE2, at the cell surface. The generated Fe^2+^ is then oxidized again by a multicopper ferroxidase (Fet3) coupled to a permease (Ftr1), which transports the Fe^3+^ ion directly into the cell [13]. The oxidase-permease system is conserved in the human pathogens *A. fumigatus* [14], *Candida albicans* [15,16] and *Cryptococcus neoformans* [17,18], as well as in the plant pathogens *Ustilago maydis* [19] and *Fusarium graminearum* [20].

Siderophore production and the RIA system are also conserved in the back yeast-like *Aureobasidium* species [21]. These ascomycetes are usually found in freshwater habitats, they can inhabit extreme environments and some species are considered opportunistic human pathogens [22]. Hydroxamate and carboxylate siderophores are produced by various fungal species derived from marine environments since the concentration of iron in seawater is extremely low [23]. In fact, the *Aureobasidium pullulans* HN6.2 strain, isolated from a marine environment, is able to produce a huge amount of fusigen, a hydroxamate siderophore [24,25].

Here, we describe the response of *Cladophialophora carrionii*, one of the chromoblastomycosis (CBM) etiological agents, to low-iron conditions. CBM is a chronic infection of cutaneous and subcutaneous tissues caused by dematiaceous fungi worldwide. *C. carrionii* is predominant in semiarid areas, while *Fonsecaea pedrosoi* is the most common agent in humid climates [26]. In 2017, CBM was included in the list of neglected tropical diseases by the World Health Organization [27,28]. CBM is a recalcitrant disease, that mainly affects individuals in poverty. The administration of antifungals, superficial surgeries or a combination of both are the therapeutic options since a standard treatment is still unavailable [29].

Like other pathogenic fungi, *Cladophialophora* developed mechanisms that enable it to establish and cause infections in host organisms. These include thermotolerance, the transition of hyphae and conidia to muriform cells and melanin production [30,31,32]. Despite nutrient acquisition, including siderophore production, being suggested as an important factor for the adaptation to the human host in black yeasts [33], investigations on such matter are still scarce. Employing in silico and experimental analyses, we demonstrated that siderophore and RIA pathways are conserved in *C. carrionii* and that iron deprivation induces both systems in this fungus. Considering the low iron availability imposed by the nutritional immunity, those high-affinity iron acquisition strategies may be relevant for pathogenesis.

## 2. Materials and Methods

### 2.1. Strain and Growth Conditions

All experiments were conducted with the *C. carrionii* KSF strain. The fungus was maintained in Potato Dextrose Agar (PDA) at 22 °C for seven days. Conidia were produced by inoculating mycelium fragments into Potato Dextrose Broth (PDB; 20% potato infusion, 2% glucose), under agitation at 150 rpm, at 28 °C, for seven days. Then, the conidia enrichment was achieved through the glass wool filtering of cultures [34]. The filtrate was centrifuged at 3000× *g* for 5 min, and conidia were resuspended in saline (0.9% NaCl). The cell suspensions were quantified in a Neubauer chamber.

For gene expression analysis experiments, mycelia grown in PDB for seven days were collected via centrifugation and washed four times with phosphate-buffered saline solution (PBS; 1.4 mM KH_2_PO_4_, 8 mM Na_2_HPO_4_, 140 mM NaCl, 2.7 mM KCl; pH 7.4). Then, fungal cells were inoculated in chemically defined medium MMcM [35], supplemented with 50 µM of the iron chelator bathophenanthroline-disulfonic acid (BPS; B-1375 Sigma-Aldrich, St. Louis, MO, USA) or 100 µM (NH_4_)_2_Fe(SO_4_)_2_ for 24 h at 28 °C. Cultures were performed in triplicates.

### 2.2. In Silico Analysis of Siderophore and RIA-Related Sequences

The search for genes related to siderophore acquisition, the RIA system and transcriptional regulation was based on sequence similarity defined using the protein BLAST tool at FungiDB (https://fungidb.org/fungidb/app/workspace/blast/new accessed on 15 January 2021). Amino acid sequences of *A. fumigatus*, *Aspergillus nidulans*, *S. cerevisiae* and *C. neoformans* were used as input in the analysis (sequences IDs are found in Appendix A). Expectation values less than or equal to 10^−10^ were considered the cut-off. The definition of orthologs was also considered based on the orthology analysis at FungiDB. Transmembrane segments of Ftr and Fet orthologs were identified using Protter (https://wlab.ethz.ch/protter/start/ accessed on 15 January 2021) [36]. Conserved motifs and amino acid residues in protein sequences were inspected manually.

The DNA Pattern Find tool from Sequence Manipulation Suite (http://www.bioinformatics.org/sms2/dna_pattern.html accessed on 15 January 2021) was employed for the identification of consensus sequences in the promoters of genes. For that, 1500 bp regions upstream the start codon of *hapX*, *sreA*, *sidI*, *sidA1*, *ftrA1* and *ftrA2* were used. Protein sequence alignments were obtained with MAFFT (https://mafft.cbrc.jp/alignment/server/ accessed on 15 January 2021), and the IQ-TREE web service (http://iqtree.cibiv.univie.ac.at/ accessed on 15 January 2021) was used for construction of the siderophore transporter phylogenetic tree [37].

### 2.3. RNA Isolation and Quantitative Real-Time PCR (qRT-PCR) Analysis

Cells grown in iron-deprived (50 μM BPS) and control [100 μM (NH_4_)_2_Fe(SO_4_)_2_] conditions were submitted to total RNA extraction. The lysis was carried out with trizol (TRI Reagent^®^, Sigma-Aldrich, St. Louis, MO, USA) and 0.5 µm glass beads in a mechanical beater (Mini-Beadbeater, Biospec Products Inc., Bartlesville, OK, USA) according to the manufacturer’s instructions. To avoid amplification due to genomic DNA, RNA was treated with DNAse I (Promega Corporation, Madison, WI, USA) and then used for cDNA synthesis (High-Capacity cDNA Reverse Transcription Kits, AppliedBiosystems Inc., Vilnius, Lithuania). The cDNA was used in qRT-PCR assays using SYBR Green PCR Master Mix (Life Technologies, Foster City, CA, USA) in the StepOnePlus^TM^ system (Applied Biosystems Carlsbad, CA, USA). The reaction conditions were as follows: 40 cycles (95 °C for 15 s/60 °C for 1 min), after initial denaturation at 95 °C for 10 min. The mRNA quantification was conducted in triplicate. The relative expression levels of transcripts were defined using the relative standard curve method [38]. The expression of the genes of interest were normalized with transcript levels of the actin gene (CLCR_02925). A melting curve analysis confirmed the primer specificity. Appendix A lists the oligonucleotides used. A student’s *t*-test was conducted to compare the expression levels of two conditions (*p* ≤ 0.05).

### 2.4. Overlay-Chrome Azurol S (O-CAS) Assay

The O-CAS assay [39] was used to detect siderophores secreted by *C. carrioni*. Solid 10-day-old MMcM mycelium cultures under iron-deprivation (without iron supplementation) and in the presence of iron [100 μM (NH_4_)_2_Fe(SO_4_)_2_] were prepared. The cultures were covered with 15 mL of O-CAS solution, monitored for changes in color and then photographed. All the reagents necessary for the CAS solution were purchased from Sigma-Aldrich, St. Louis, MO, USA. Residual traces of iron were removed from glassware with acid treatment [40].

### 2.5. Ferric Perchlorate Assay

Supernatants from *C. carrioni* 30-day-old cultures under iron deprivation (without iron supplementation) and control conditions [100 μM (NH_4_)_2_Fe(SO_4_)_2_] were submitted to the ferric perchlorate colorimetric assay [41]. The samples were filtered (0.22 μM pore membrane) and lyophilized. Dried supernatants were resuspended in water (MilliQ-water) using one-tenth of the original volume. For the assay, 250 μL of samples was added to 1.250 mL of 5 mM Fe(ClO_4_)_3_. The same volumes of sterile MMcM with and without iron were used as controls. The samples were incubated at room temperature for 5 min and monitored for a change in color.

### 2.6. Siderophore Identification

To induce siderophore production, *C. carrionii* mycelia were cultivated in an iron-free MMcM medium for 30 days. Culture supernatants were collected and filtered in 0.22 µM membranes in order to remove any fungal cells. Samples were freeze-dried and then resuspended in one-tenth of the initial volume. These samples were used for extracellular siderophore identification. For intracellular siderophore identification, cells were collected via centrifugation at 3000× *g* and 4 °C for 15 min and washed 5 times with PBS to remove extracellular siderophores. Fungal cells were lysed through mechanical disruption in a mortar, and the samples were kept frozen by adding liquid nitrogen. The lysate containing the intracellular siderophores was resuspended in sterile water (1 mL/4 mL of culture). Cell debris was removed via centrifugation, and the obtained supernatant was filtered in 0.22 µM and lyophilized. For identification assays, the lyophilized material was resuspended in one-tenth of the original volume in MilliQ water.

For siderophore identification, 100 mM FeSO_4_ was firstly added to the samples. Then, the siderophores were purified with an Amberlite XAD-16 column (Rohom and Haas, Philadelphia, PA, USA). Elution was performed with methanol, and the eluates were dried under a vacuum. Then, 100 µL of water was used for pellet solubilization; a 10 µL aliquot was loaded onto the HPLC containing a reversed-phase column (Nucleosil C18, 5 mm, 4 × 250 mm, Macherey-Nagel, Duren, Germany). The chromatography was performed as described [42]. The collected fractions were vacuum-dried, resuspended in a 50 methanol/0.1% formic acid solution, and analyzed in the LTQ Velos ion trap mass spectrometer (Thermo Fisher Scientific Waltham, MA, USA) equipped with an electrospray source (ESI-MS, Electrospray Ionization Mass Spectrometry). The electrospray source parameters were 4.0 kV and 270 °C.

## 3. Results

### 3.1. Genes for High-Affinity Iron Acquisition Are Conserved in C. carrionii

As a first step in the investigation of iron acquisition mechanisms in *C. carrionii*, an orthology analysis was conducted. Genes of *A. fumigatus*, *S. cerevisiae*, *A. nidulans* and *C. neoformans* were used in the search (Appendix A). All genes required for the synthesis of hydroxamate siderophores were identified in the *C. carrionii* genome (Table 1), with the exception of *sidG*, involved in TAFC biosynthesis in *A. fumigatus*. A genome location inspection indicated that genes are localized near each other (Figure 1A). Two ornithine monooxygenase orthologs were identified in *C. carrionii*. One of them, described here as *sidA1*, is located in and shares the promoter region with NRPS *sidC*. A second gene cluster includes five orthologs involved in siderophore biosynthesis: NRPS *sidD*, transacylase *sidF*, enoyl-CoA hydratase *sidH*, acyl-CoA synthetase *sidI* and ornithine monooxygenase (described here as *sidA2*). Interestingly, this cluster also harbors a putative ABC multidrug transporter (CLCR_02231).

Considering that *C. carrionii* can synthesize siderophores, according to the genome analysis, we next searched for the presence of siderophore transporters. *A. fumigatus*, *A. nidulans*, *S. cerevisiae* and *C. neoformans* amino acid sequences were used in the similarity analysis (Appendix A). *C. carrionii* presents six putative siderophore transporters (Table 1), one of which is localized near the *sidA2* in the siderophore biosynthetic cluster (CLCR_02321, Figure 1A). The number of transmembrane domains (TMs) varies from 13 to 14, which is in accordance with the predicted TMs (12–14) in MFS members [9]. Phylogeny analysis suggested that *C. carrionii* carries orthologs to MirB (CLCR_03365), MirD (CLCR_02321), Sit1 (CLCR_01135) and Sit2 (CLCR_06890 and CLCR_03414) (Figure 2). The *C. carrionii* transporter CLCR_04614 did not cluster with any of the analyzed sequences. A deeper investigation revealed that CLCR_04614 shares a higher sequence similarity (E-value 3 × 10^−156^) with Afu3g01360, an *A. fumigatus* MFS transporter for which the function is not well defined [43].

In addition to siderophore production and uptake, the RIA system allows for high-affinity iron acquisition in fungi. This system comprises three groups of proteins: ferric reductases (Fres), multicopper ferroxidases (Fets) and high-affinity iron permeases (Ftrs). Fres are characterized by the presence of three conserved domains: a ferric reductase domain (FRD) in the transmembrane regions, a FAD-binding and NADPH-binding domains. The last two are localized in the C-terminal cytoplasmic region of Fres [44]. The FRD domain, predicted using the Pfam database (PF01794), was searched against the *C. carrionii* amino acid sequence database in FungiDB (https://fungidb.org/fungidb/app/, accessed on 15 January 2021). Twelve sequences were retrieved, eleven of which contain both the FAD-binding (PF08022/IRP017927) and the NADPH-binding domains (PF08030) (Table 2). The number of transmembrane domains varies from five to eight, and three sequences have predicted signal peptides.

Based on similarity analysis with *S. cerevisiae* and *A. fumigatus* sequences (Appendix A), we found that *C. carrionii* carries two orthologs of the iron permease Ftr1/FtrA and two orthologs of the ferroxidase Fet3/FetC (Table 1). *C. carrionii* sequences were defined as FtrA1, FtrA2, FetC1 and FetC2. Each member of the permease–ferroxidase pairs, FtrA1-FetC1 and FtrA2-FetC2, is localized next to each other in the genome and shares the same promoter (Figure 1B). Iron permeation by the Ftr1 depends on specific motifs, including the elements REXLE and DXXE [54,55]. Both were identified in the amino acid sequences of the two putative *C. carrionii* permeases (CLCR_07299 and CLCR_03583) (Figure 3A). The REXLE element is located in the transmembrane domains 1 and 4, while the DXXE domain is found in the extracellular loop between transmembrane domains 6 and 7. Additionally, the topology of putative *C. carrionii* Ftrs is similar to that found in *S. cerevisiae* and *C. albicans* [54,55], presenting seven transmembrane domains with an extracellular N-terminal and intracellular C-terminal (Appendix A).

Ferric iron transported by Ftr1 results from the action of Fet3. This is a type 1 membrane protein that contains only one transmembrane domain. The N-terminus is extracellularly localized while the C-terminus is cytoplasmic [56,57]. As an MCO, Fet3 presents three types of copper-binding sites that allow for its ferroxidation activity [58]. The two putative *C. carrionii* ferroxidases (CLCR_03382 and CLCR_06747) present the same topology as *S. cerevisiae* Fet3, with three MCO domains localized in the extracellular N-terminus (Appendix A). In addition to the copper-binding sites, Fet3 has some specific amino acid residues involved in (I) Fe^2+^ binding, (II) the coupling of electron transfer and/or (III) Fe^3+^ transfer to Ftr1. Those residues include E185, D278, D283, Y354 and D409. E185 and D409 have been demonstrated to play a role in all three processes [58]. Y354 and D278 is necessary for the ferroxidase activity of Fet3 [59,60], and D283 is related to the trafficking of Fe^3+^ to Ftr1 [58]. Sequence alignment of *S. cerevisiae* Fet3, *A. fumigatus* FetC and the predicted Fets of *C. carrionii* demonstrated that all the above-mentioned amino acid residues are conserved in this black fungus (Figure 3B). Altogether, these findings strongly suggest that *C. carrionii* may acquire iron via the RIA system.

Siderophore-based iron acquisition, as well as the RIA system, is regulated by iron availability via SreA and HapX transcription factors, for which orthologs were also identified in the *C. carrionii* genome (Table 1 and Appendix A). SreA is a GATA-type regulator that represses high-affinity iron uptake under iron sufficiency, while the bZIP factor HapX induces RIA and siderophore uptake in low iron availability [61,62]. Taken together, our in silico approach demonstrated that the machinery responsible for high-affinity iron uptake is conserved in this black fungus.

### 3.2. Iron Availability Regulates the Expression of Iron Acquisition Genes

Genes related to iron acquisition are usually regulated by the availability of this metal in the extracellular environment. To test this hypothesis in *C. carrionii*, the expression of selected genes was determined under iron-deprived conditions. As shown in Figure 4A, *sidA1* and *sidI*, representative of siderophore biosynthesis, as well as *ftrA1* and *ftrA2*, involved in RIA, were induced in the presence of the iron chelator BPS. The transcription factors *sreA* and *hapX* were regulated by iron as well. *sreA*, known to repress the iron acquisition system in iron-sufficient conditions, was repressed under iron starvation, while *hapX*, an inducer of iron acquisition, was positively regulated in BPS, as expected.

An inspection of the promoter regions of the regulated genes (Figure 4B) revealed the presence of consensus sequences recognized by SreA (HGATAR, and its extended version ATCWGATAA) [42,61,63] and HapX (CCAAT) [64], indicating that both transcription factors may regulate iron acquisition in *C. carrionii*. *sidA1* shares the 5′ upstream region with the putative NRPS SidC (CLCR_04490), while the promoters of the permeases *ftrA1* and *ftrA2* are shared with *fetC1* and *fetC2*, respectively (Figure 1). Therefore, we can infer that *sidC*, *fetC1* and *fetC2* are regulated by iron availability as well.

### 3.3. C. carrionii Produces and Secretes Hydroxamate Siderophores under Low-Iron Conditions

Based on the genomic information that *C. carrionii* is capable of synthesizing siderophores, this phenotype was investigated. Cells were incubated for 10 days in solid MMcM medium supplemented (100 μM) or not with iron. Next, the plates were overlaid with chrome azul S (O-CAS), a blue solution for which color is changed in the presence of siderophores since these molecules “steal” the Fe ion from CAS. As shown in Figure 5A, in 15-day-old cultures, CAS changed from blue to orange in low-iron medium. This indicates that siderophores were produced and secreted by *C. carrionii*. As expected, the presence of iron blocked siderophore biosynthesis.

The development of an orange color from CAS solution indicates the presence of hydroxamate-type siderophores [39]. To confirm the nature of siderophores produced by *C. carrionii*, the ferric perchlorate assay was performed. A Fe(ClO_4_)_3_ solution was mixed with concentrated supernatants of 30-day cultures in low-iron medium. The appearance of an orange-red color in the −Fe supernatant revealed the presence of hydroxamates (Figure 5B). As expected, the +Fe condition inhibited the production of siderophores. Non-inoculated MMcM mixed with Fe(ClO_4_)_3_ is shown as an additional control.

### 3.4. Ferricrocin Is Produced by C. carrionii as an Intra- and Extracellular Siderophore

Aiming for the identification of the hydroxamate siderophores secreted by *C. carrionii*, culture supernatants were submitted to RP-HPLC and mass spectrometry analysis. Following incubation in iron-deprived MMcM for 30 days, the culture supernatant was filter-sterilized and concentrated ten times with ultrapure water. FeSO_4_ was then added to the sample in order to saturate the secreted siderophores. An analysis via RP-HPLC demonstrated the presence of a compound displaying absorption at 430 nm, typical of iron-saturated siderophores. The retention time at which the peak was observed matched that of ferricrocin, a hydroxamate siderophore usually utilized for iron storage (Figure 6). To confirm the siderophore identity, the RP-HPLC peak was subjected to high-resolution mass spectrometry. The presence of ferricrocin (C_28_H_47_N_9_O_13_Fe) was reinforced through the visualization of four different ionizing adducts: H^+^ (*m*/*z* = 771.2471), NH_4_^+^ (*m*/*z* = 788.2733), Na^+^ (*m*/*z* = 793.2279) and K^+^ (*m*/*z* = 809.2021) (Appendix A).

Next, the presence of intracellular siderophores was evaluated in cell extracts obtained after incubation for 30 days in iron-free MMcM. Following centrifugation, cells were washed five times with PBS to eliminate extracellular siderophores and processed as described in the Materials and Methods. RP-HPLC analysis was conducted after saturation of the sample with FeSO_4_, and a peak compatible with ferricrocin was again identified (Figure 6, third panel) and confirmed via high-resolution mass spectrometry.

## 4. Discussion

Although the importance of siderophore production by black yeasts in the adaptation to the human host was cited a long time ago [33], the molecular strategies used by those fungi for iron acquisition are still poorly explored. Here, we showed that *C. carrionii* KSF expresses two high-affinity iron-acquisition systems, siderophore production and RIA, under iron limiting conditions.

Firstly, a homology analysis was carried out to search for gene orthologs to those involved in iron assimilation in other fungi. The machinery for the biosynthesis and uptake of siderophores was identified. Most of the genes are clustered, a characteristic hallmark of secondary metabolite production, found in various fungal classes belonging to basidiomycetes and the Ascomycota phylum [42,65,66]. This arrangement enables the genes to be co-regulated. *C. carrionii* presents two *sidA* orthologs, a common feature in some plant and human pathogens [9], including the closely related *Cladophialophora bantiana* [66]. Genomic analyses of *C. bantiana* isolated from a brain abscess revealed the presence of two clusters of siderophore biosynthesis genes. One includes *sidA* and *sidC* orthologs and the other is represented by *sidD*, *sidF*, *sidI*, *sidA* and a siderophore transporter [66]. In *C. carrionii*, we found the same pattern of gene distribution into two clusters, with the addition of *sidH*, which shares the 5′ region with *sidF*. Two siderophore biosynthetic clusters were also identified in the black yeast-like *Aureobasidium melanogenum*, one including *sidA* and *sidC* orthologs and the other composed by *sidD*, *sidF* and *sidI* [21].

An orthology survey indicated six putative siderophore transporters in the *C. carrionii* genome. Phylogenetic analysis, however, showed that CLCR_04614 is not closely related to a siderophore transporter with a proven function in other fungi. In contrast, it seems to be more similar to an uncharacterized *A. fumigatus* MFS transporter [43]. The number of siderophore transporter paralogs in *C. carrionii* is equivalent to that in *A. fumigatus*, which encodes five transporters [61], and the basidiomycete *C. neoformans*, which presents six [9]. CLCR_02321 is localized within the larger cluster found in *C. carrionii* and shares its promoter region with the ornithine monooxygenase-encoding gene *sidA2*. The presence of a siderophore transporter in a siderophore biosynthesis cluster is a feature conserved in other ascomycetes, including *C. bantiana* [66], *Blastomyces dermatitidis* [67], *A. fumigatus* [61], *Histoplasma capsulatum* [63], *A. melanogenum* [21] and the *Paracoccidioides* genus [42].

The recognition of siderophores by the plasma membrane transporters is highly stereospecific [68]. *C. carrionii* presents six putative siderophore transporters, five of which (MirB, MirD, Sit1 and Sit2) are orthologs to *A. fumigatus* proteins for which substrate specificity has been elucidated. In *A. fumigatus*, MirB and MirD transport TAFC and fusarinine C, respectively [43], while Sit1 and Sit2 have overlapping and unique substrate specificities. Both Sit1 and Sit2 mediate the transport of ferrichrome, ferricrocin and coprogen-type siderophores, whereas only Sit1 transports the bacterial ferroxiamines [69]. The last are considered xenosiderophores, that is, those produced by other microorganisms. Considering the diversity of siderophore transporters and putative specificities in *C. carrionii*, we may infer that this fungus is able to take up a variety of molecules, including its native and the xenosiderophores. In fact, various fungal species that lack the siderophore synthesis machinery, like *C. albicans*, *Candida glabrata, S. cerevisiae* and *C. neoformans*, have siderophores transporters for the uptake of xenosiderophores [70].

In the environment, *C. carrionii* can be found in soil, where ferrioxamines, ferricrocin and ferrichromes are abundantly found [71], decaying matter, wood and spines and plants, mainly in *Cactaceae* [72,73]. In an analysis of cactus microbiomes, Fonseca-Garcia and Coleman-Derr [74] identified bacteria able to produce siderophores. These facts reinforce the hypothesis that *C. carrionii* could be able to use xenosiderophores in the environment. Additionally, secondary bacterial infections are the most frequent complications of CBM [75] and siderophores produced by these prokaryotes may be a source of iron for *C. carrionii* in the mammalian host.

Beyond siderophore synthesis and uptake, the RIA system is also conserved in *C. carrionii.* Twelve sequences containing the FRD domains, a hallmark of ferric reductases, were identified in the *C. carrionii* genome. Most of them belong to ortholog groups that harbor characterized Fres in other fungi. *S. cerevisiae* has eight FRE orthologs, FRE1-FRE8. FRE1 and FRE2 are involved in both ferric and cupric ion reductions [45,46,76], FRE3-FRE6 are regulated specifically by iron, while FRE7 is specifically regulated by copper [77]. FRE8 is related to both iron and copper homeostasis [53]. Additionally, FRE3 and FRE4 are required for the uptake of Fe^3+^ bound to siderophores via the reduction of this ion to Fe^2+^ [78]. The several Fre orthologs in *C. carrionii* may correlate with diversity in function, i.e., the reduction of free iron and copper, as well as siderophore-bound iron. Further investigations are necessary to test this hypothesis.

Two orthologs of Ftr1/FtrA and Fet3/FetC were identified in *C. carrionii*. FtrA and FetC are divergently transcribed from the same intergenic region, a characteristic conserved in many fungal divisions [79], including the black yeast-like *A. melanogenum* [21]. Three ferroxidases and two iron permeases were found in the genome of *C. bantiana* isolated from the brain abscess of a patient. Two of the Fets are located next to the two Ftrs [66]. *C. carrionii ftrA1* and *ftrA2* are up-regulated upon iron deprivation. As both share an intergenic region with *fetC1* and *fetC2*, respectively, we assume that the oxidases are also regulated by iron availability. The co-expression of Ftr-Fet occurs in many fungal species, including *A. fumigatus* [14] and the mucormycosis agent *Lichtheimia corymbifera* [79]. Indeed, in *S. cerevisiae* and *C. albicans*, the co-expression of Ftr1 is required for the trafficking of Fet3 to the plasma membrane and the efficient localization of Frt1 at the cell surface depends on the co-expression of Fet [55,80].

The adaptation to iron availability in *A. fumigatus* is regulated by SreA and HapX [61,62]. This transcriptional control is conserved in fungi, and the functions of SreA and/or HapX orthologs have been described in model fungal species, as well as in human and plant pathogens [81,82,83,84,85,86,87]. In most pathogenic species, these transcription factors are virulence determinants. Here, we demonstrated that iron availability regulates *sreA* and *hapx* expression in an opposite way. While *sreA* is repressed, *hapX* is induced by iron starvation. This agrees with the postulated negative feed-back loop that connects these two proteins in *A. fumigatus*: SreA is induced under iron sufficiency and represses *hapX*, whereas HapX is positively regulated by iron limitation and represses *sreA*. The functional connection of SreA and HapX in *C. carrionii* is reinforced by the presence of regulatory sequences in their promoter regions. Siderophore biosynthesis (*sidA1* and *sidI*)- and RIA (*ftrA1* and *ftrA2*)-related genes were up-regulated by iron starvation, probably via the action of HapX. *A. fumigatus sidA* mutants are unable to grow upon iron deprivation and are avirulent in a mouse model of invasive aspergillosis [14,88], and *sidI* deletion also impacts growth under low iron in this mold [11]. *Aureobasidium* species present SreA and HapX orthologs that are regulated by iron, as in *A. fumigatus*. Additionally, siderophore biosynthesis genes harbor SreA and HapX consensus sequences in their promoter regions, confirming the role of these two transcription factors in the regulation of iron homeostasis in these black yeast-like strains [21,89]. Indeed, *sidA* expression was derepressed under iron sufficiency in an Sre1-knockout *A. pullulans* strain [89].

The induction of siderophore and RIA acquisition systems was also reported in the black yeast *Exophiala dermatitidis* recovered from an ex vivo skin model of infection [90]. This fact confirms that iron limitation is faced by implantation pathogens, like *C. carrionii*, when in contact with the host. A *C. albicans* strain defective in a siderophore transporter is less efficient in the invasion of keratinocyte layers, which demonstrates the fundamental role of siderophore uptake in invasion and the penetration of epithelial tissue [91]. In addition, the lack of a functional RIA system makes *C. albicans* and *C. neoformans* attenuated for virulence [15,17,18] A comparative genomic survey of pathogenic and environmental siblings of *Cladophialophora* and the *Fonsecaea* genus identified an enrichment of genes related to siderophore biosynthesis and uptake in the clinical species. This suggests a role of these cellular processes in virulence since no enrichment was observed in the environmental species [92].

Considering that the siderophore biosynthesis pathway is up-regulated upon iron deficiency, the production of hydroxamates was evaluated in vitro. The O-CAS and ferric perchlorate assays confirmed that *C. carrionii* produces and secretes hydroxamate-type siderophores. This class includes four families with distinct structures: fusarinines, coprogens, ferrichromes and rhodotorulic acid [9]. An analysis of supernatants of iron-deprived cultures revealed the presence of ferricrocin, a typical intracellular siderophore that belongs to the ferrichrome family, of which the synthesis depends on SidC (CLCR_04490 in *C. carrionii*) [93]. Ferricrocin is used as an iron-storage compound in *Aspergillus* species and *Neurospora crassa* [93,94] and is involved in intra- and transcellular iron distribution during conidiogenesis in *A. fumigatus* [95]. By contrast, in the plant pathogen *Magnaporthe grisea*, ferricrocin is dispensable for conidiation but is essential for pathogenesis [96]. A lack of ferricrocin also causes attenuation of virulence in a murine aspergillosis model [10]. Indeed, it was recently demonstrated that in *A. fumigatus*, ferricrocin contributes to iron acquisition as an extracellular siderophore, in addition to the intracellular role described previously [97]. The black yeast *Aureobasidium melanogenum* HN6.2 strain produces ferricrocin as an intra- and extracellular siderophore as well [21]. Sit1 and Sit2 of *A. fumigatus* [69] and Sit1 of *C. albicans* [91] are the membrane transporters responsible that mediate ferricrocin uptake. The presence of orthologs to those proteins in *C. carrionii* suggests that secreted ferricrocin is further utilized by the fungus. As expected, the presence of ferricrocin was confirmed in *C. carrionii* cellular extracts as well.

In summary, our data demonstrate that *C. carrionii* employs two high-affinity iron uptake systems (Figure 7). The siderophore-mediated acquisition comprises enzymes involved in hydroxamate biosynthesis and plasma membrane siderophore transporters. The RIA pathway includes ferric reductases, multicopper ferroxidases and high-affinity iron permeases. Representative genes for each pathway are induced by iron starvation, and this regulation is probably accomplished by the activities of HapX and SreA transcription factors.

## 5. Conclusions

To the best of our knowledge, this is the first report of iron acquisition systems in the *Cladophialophora* genus. This black fungus encodes the genomic machinery for siderophore production and RIA systems, which are induced by iron starvation in vitro. In such conditions, the hydroxamate siderophore ferricrocin is found in cell extracts and is also secreted, as recently demonstrated in *A. fumigatus*. *C. carrionii* codifies six putative siderophore transporters suggesting a versatility in the uptake of different types of siderophores. In addition, the RIA system may be a contributor in the acquisition of siderophore-bound iron. Siderophores are essential for virulence in various plant and human fungal pathogens. As such, these molecules have been investigated as potential biomarkers of fungal infections and as Trojan horses for the oriented delivery of antimicrobials, since SITs are fungal-specific transporters. Furthermore, siderophore biosynthesis is not found in humans, making this pathway a target for drugs. Finally, the scarce number of investigations focusing on molecular mechanisms of virulence and pathogenicity of CBM agents confirms the neglected nature of this disease. Thus, efforts must be made in order to gather information on such matters to open new possibilities of more effective treatments. This meets some objectives of the 2030 Agenda of the United Nations, in which the relevance of neglected tropical diseases is highlighted.

## Figures and Tables

**Figure 1 jof-09-00727-f001:**
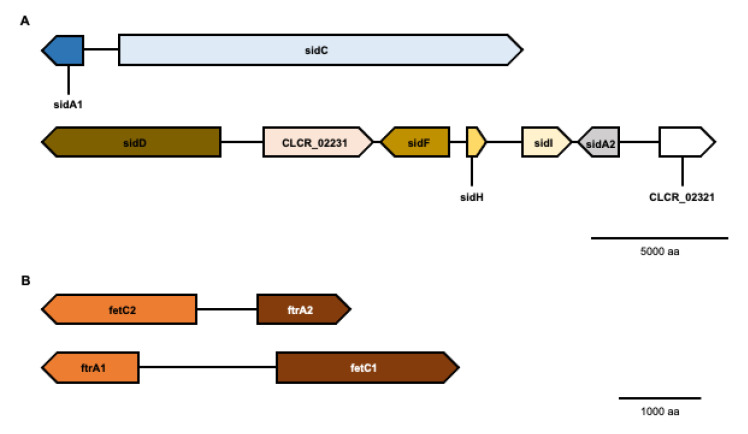
Genomic organization of siderophore and RIA systems in *C. carrionii*. (**A**) Genes related to hydroxamate synthesis and transport (CLCR_02321) are organized in two clusters, each one containing a *sidA* ortholog. (**B**) Two ferroxidases (FetC1 and FetC2) and two iron permeases (FtrA1 and FtrA2) are encoded by *C. carrionii*. The permease–oxidase pairs share the 5′ upstream region.

**Figure 2 jof-09-00727-f002:**
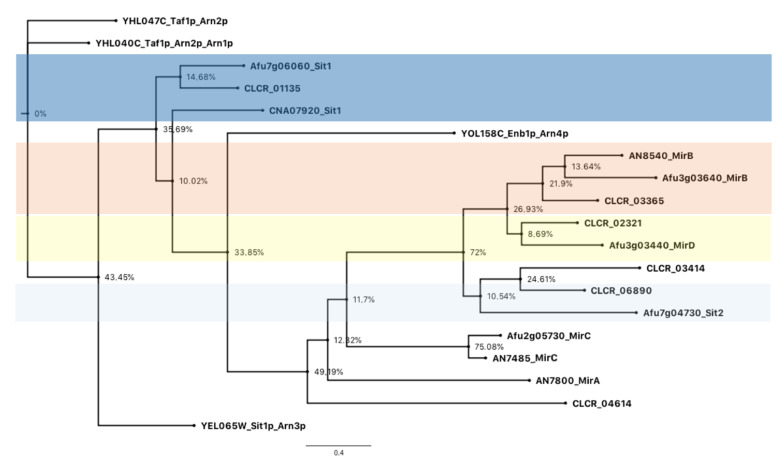
Phylogenetic analysis of *C. carrionii* siderophore transporters. The six putative SITs of *C. carrionii* (CLCR_) were grouped with *A. fumigatus* (Afu), *A. nidulans* (AN) and *C. neoformans* (CN) transporters in different clusters, suggesting the presence of MirB, MirD, Sit1 and Sit2 orthologs in *C. carrionii*.

**Figure 3 jof-09-00727-f003:**
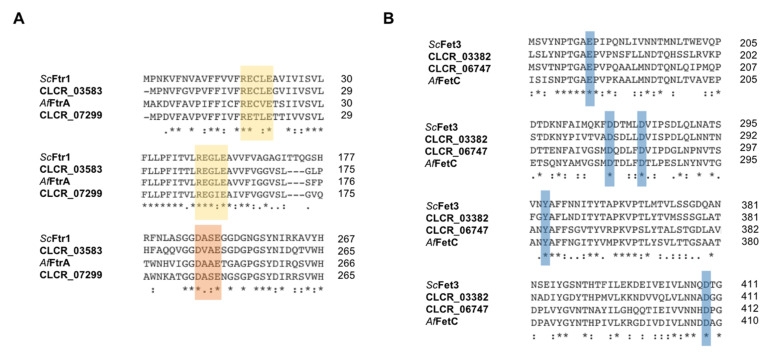
Structural features of the putative iron oxidase/permease complex in *C. carrionii*. Ftrs and Fets structural features required for iron trafficking are shown. (**A**) Alignment of putative *C. carrioni* Ftr1 orthologs (CLCR_03583 and CLCR_07299) with REXLE and DXXE motifs highlighted in yellow and orange, respectively. *Sc*Ftr1: *S. cerevisiae* YER145C, *Af*FtrA: *A. fumigatus* Afu5g03800. (**B**) Alignment of predicted *C. carrioni* Fet3 orthologs (CLCR_03382 and CLCR_06747) showing essential amino acids residues in blue (in *S. cerevisiae* Fet3: E185, D278, D283, Y354 and D409 residues; in CLCR_03382: E182, D275, D280, Y354 and D409 residues; in CLCR_06747: E187, D280, D285, Y355 and D410 residues). *Sc*Fet3: *S. cerevisiae* YMR058W, *Af*FetC: *A. fumigatus* Afu5g03790. ***** represents conserved amino acids and : represents semiconserved amino acids.

**Figure 4 jof-09-00727-f004:**
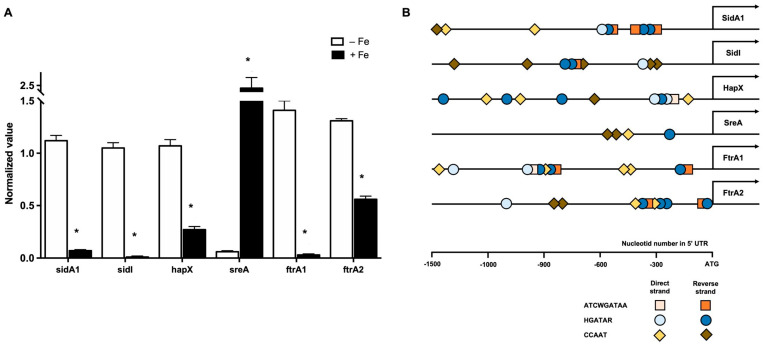
Expression profile (**A**) and consensus sequence determination (**B**) of high-affinity iron acquisition genes. (**A**) Transcript levels of siderophore biosynthesis (*sidA1* and *sidI*), RIA (*ftrA1* and *ftrA2*) and transcriptional regulator (*hapX* and *sreA*) genes under low-iron conditions (50 μM BPS, −Fe). Actin transcripts were used as the endogenous control. Statistically significant difference between −Fe and +Fe conditions was determined based on a Student’s *t*-test (* *p* < 0.05). qRT-PCR was performed with samples from triplicates. (**B**) Consensus sequences for SreA (HGATAR and ATCWGATAA) and HapX (CCAAT) were search within 1500 nt upstream of the predicted start codons of shown genes. Both direct and reverse strands were considered. H: A/T/C, W: A/T, R: A/G.

**Figure 5 jof-09-00727-f005:**
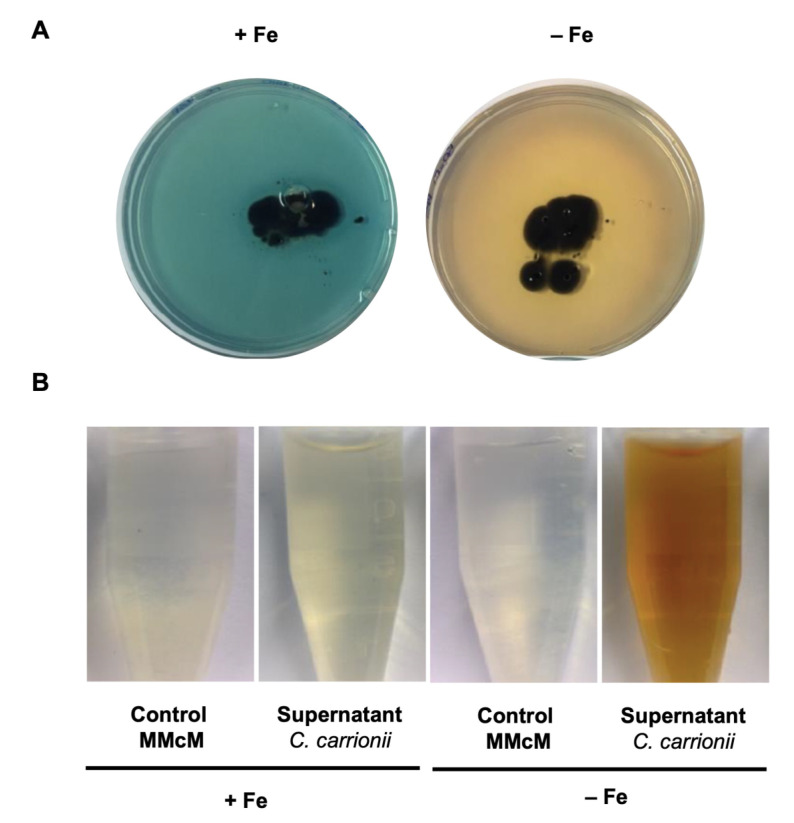
Detection of hydroxamate siderophores in low-iron medium. (**A**) *C. carrionii* secreted siderophores following growth on MMcM agar plates deprived of iron. The presence of hydroxamates is indicated by the change of CAS from blue to orange. In the presence of iron (100 μM), siderophore production is abrogated. (**B**) The nature of the secreted siderophores was confirmed through the ferric perchlorate assay. Supernatants of iron-deprived cultures presented an orange-red color after the addition of Fe(ClO_4_)_3_. Control MMcM represents the non-inoculated medium.

**Figure 6 jof-09-00727-f006:**
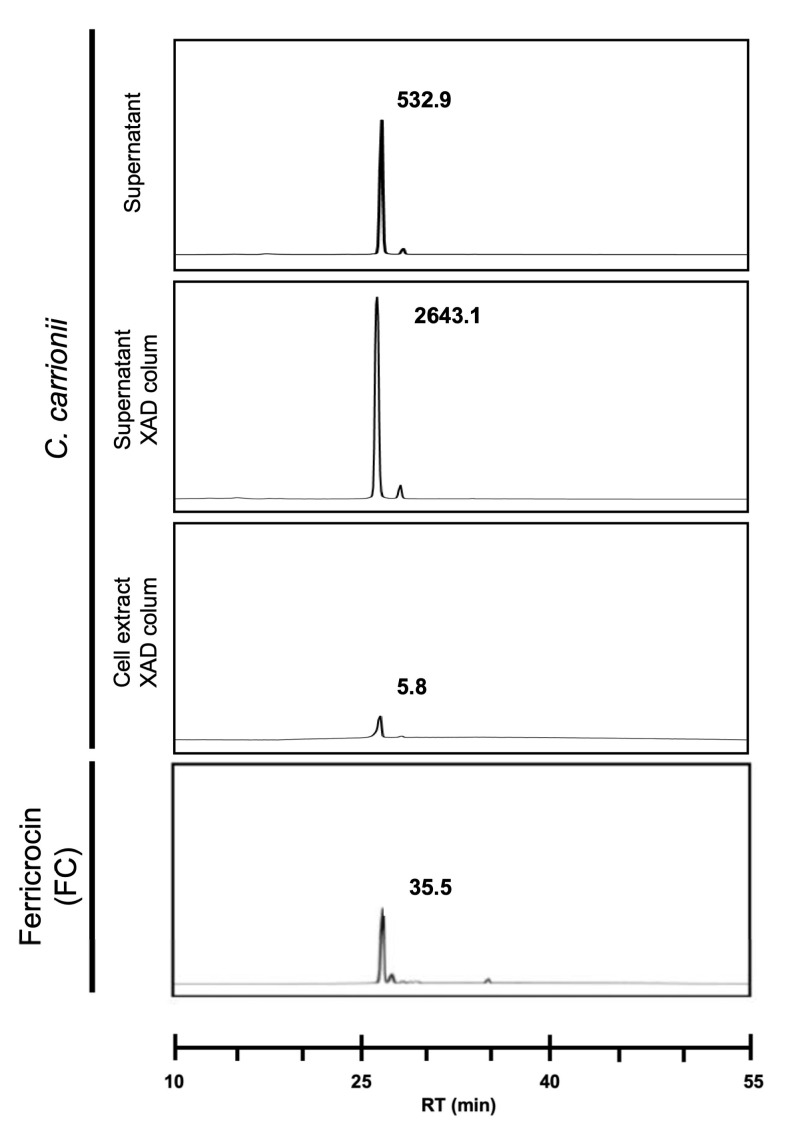
Identification of ferricrocin in *C. carrionii* culture supernatants and cell extracts. Reversed-phase HPLC of *C. carrionii* cultivated in iron-limiting conditions demonstrated the presence of ferricrocin in cell extracts and supernatants. Intra-(cell extract) and extracellular (supernatant) siderophores were purified with an XAD column. The *Y*-axis indicates the absorption at 430 nm (reddish colored substances) in arbitrary units; the units for the identified peaks is given. RT: retention time.

**Figure 7 jof-09-00727-f007:**
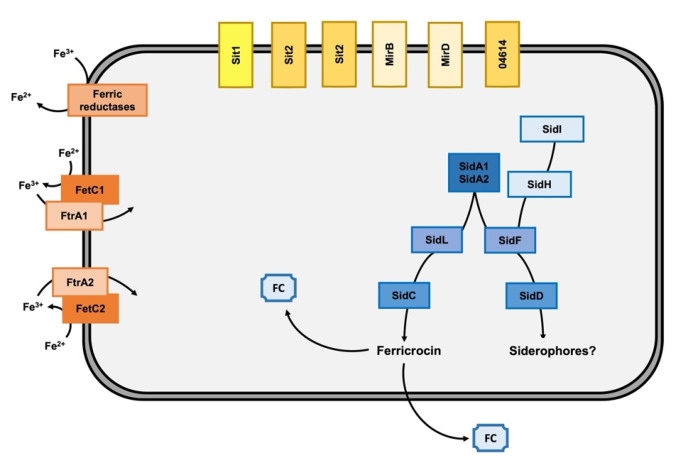
Schematic of the high-affinity iron-acquisition systems in *C. carrionii*. Siderophore biosynthesis enzymes are shown in shades of blue, and siderophore transporters are demonstrated in yellow. Components of RIA are presented by shades of orange. Ferricrocin (FC) is found intracellularly and as a secreted siderophore. The connection of siderophore biosynthesis with the ergosterol pathway is demonstrated based on SidI and SidH. Siderophores possibly synthesized by the SidF-SidD branch were not identified in this study.

**Table 1 jof-09-00727-t001:** Ortholog genes putatively involved in high-affinity iron acquisition in *C. carrionii ^#^*.

Iron Related Pathway	*Cladophialophora carrionii* Genes *
Siderophore biosynthesis	CLCR_04832 (SidA1)
	CLCR_02414 (SidA2)
	CLCR_02426 (SidF)
	CLCR_04490 (SidC)
	CLCR_02213 (SidD)
	CLCR_04002 (SidL)
	CLCR_02353 (SidI)
	CLCR_02613 (SidH)
Siderophore transport	CLCR_02321 (MirD)
	CLCR_03365 (MirB)
	CLCR_06890 (Sit2)
	CLCR_03414 (Sit2)
	CLCR_01135 (Sit1)
	CLCR_04614
Reductive iron acquisition (RIA)	CLCR_07299 (FtrA1)
	CLCR_03583 (FtrA2)
	CLCR_06747 (FetC1)
	CLCR_03382 (FetC2)
Regulation of iron metabolism	CLCR_05818 (SreA)
	CLCR_01554 (HapX)

* Sequence IDs are according to FungiDB (https://fungidb.org/fungidb/app accessed on 15 January 2021). ^#^ Detailed results of the similarity analysis are found in Appendix A.

**Table 2 jof-09-00727-t002:** Features of putative ferric reductases encoded by *C. carrioni*.

*C. carrionii* Sequence ID	Ortholog Group	Orthologs in Fungi *	Signal Peptide	TM Domains	FRD ^#^	FAD-Binding Domain ^#^	NAD-Binding Domain ^#^
CLCR_00371	OG6_114877	FRE2*Sc*/CFL1*C*a/FRE10*Ca*/FREB*Af*	MAGLRYLGPIILVLSTIALA	7	✓	✓	✓
CLCR_00599	OG6_114877	FRE2*Sc*/CFL1*C*a/FRE10*Ca*/FREB*Af*	N/A	7	✓	✓	✓
CLCR_00776	OG6_114877	FRE2*Sc*/CFL1*C*a/FRE10*Ca*/FREB*Af*	MAYLVAVWTFCLFLARVANA	7	✓	✓	✓
CLCR_01862	OG6_137720	FRE1Sc	N/A	6	✓	✓	✓
CLCR_04640	OG6_210498	-	N/A	6	✓	✓	N/A
CLCR_04694	OG6_100242	FRE2*Cn*	N/A	6	✓	✓	✓
CLCR_06584	OG6_119573	FRP1/FRP2*Ca*	N/A	7	✓	✓	✓
CLCR_06696	OG6_114876	FRE7*Sc*	N/A	8	✓	✓	✓
CLCR_06800	OG6_151519	-	N/A	6	✓	✓	✓
CLCR_06849	OG6_142743	FRE8*Sc*	N/A	7	✓	N/A	✓
CLCR_09538	OG6_114877	FRE2*Sc*/CFL1*C*a/FRE10*Ca*/FREB*Af*	MKVLTILLTLSALSSA	6	✓	✓	✓
CLCR_11337	OG6_109931	-	N/A	5	✓	✓	✓

* The ortholog definition was based on characterized Fres in the literature: FRE1 *S. cerevisiae* [45], FRE2/CFL1/FRE10/FREB *S. cerevisiae* [46], *C. albicans* [47,48] and *A. fumigatus* [49], FRE2 *C. neoformans* [50], FRP1/FRP2 *C. albicans* [51], FRE7 *S. cerevisiae* [52] and FRE8 *S. cerevisiae* [53]. ^#^ Domains: ferric reductase-like transmembrane component (FRD, PF01794), FAD-binding domain (PF08022/IRP017927) and NADPH-binding domain (PF08030).TM: transmembrane domains, N/A: not available. Proteins were characterized based on FungiDB (https://fungidb.org/fungidb/app/workspace/blast/new accessed on 15th January 2021).

## Data Availability

Not applicable.

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
