# Peer review of "Iron Starvation Induces Ferricrocin Production and the Reductive Iron Acquisition System in the Chromoblastomycosis Agent Cladophialophora carrionii"

_jof, 2023, doi:10.3390/jof9070727_

Round 1

Reviewer 1 Report

This work focuses on the production of siderophores and the reductive iron acquisition system (RIA) employed by Cladophialophora carrionii under iron restriction. This work holds its significance as it explains the relationship between iron acquisition and survival of C. carrionii facing iron starvation. The design of this work is good, and sufficient results have been presented. I have only one suggestion about the Introduction and discussion sections.

  As we know, the iron acquisition and homeostasis have been extensively elucidated in Aspergillus fumigatus, while, recently these biological processes have also been specified in Aureobasidium spp.. More interestingly, several characteristics of C. carrionii regarding iron acquisition and homeostasis, share close similarity to those in Aureobasidium. Hence, more backgrounds and discussion are required against this yeast-like fungus.

  Due to this reason, I suggest the authors to improve this manuscript at this point.

Author Response

Response: We thank the reviewer for this valuable comment. An extensive review on Aureobasidium spp. iron acquisition mechanisms was performed and information on this regard were added in the Introduction and Discussion sections (changes highlighted in yellow).

Reviewer 2 Report

The authors first reported iron acquisition systems in the Cladophialophora genus. They found that this black fungus encodes a genomic machinery for siderophore production and RIA systems which are induced by iron starvation in vitro. The research design is good, and a variety of reliable methods were employed. The paper is well-written and provides sufficient supportive information, making it easy to follow. The presentation of data, particularly in the figures, is well-organized and includes detailed information.

And I have some other comments below:

Line 85-86: Please check your citation to make sure they are consistent throughout the manuscript.

Line 222: Could you please provide another figure with high quality?

Line 326: Can Cladophialophora produce spores? Could you apply spore suspension to do the CAS assay? Please refer to the paper published in PNAS. They use CAS assay for the detection of secreted copper-binding molecules. They ensure the spore concentrations are the same at the beginning. Raffa, Nicholas, et al. "Dual-purpose isocyanides produced by Aspergillus fumigatus contribute to cellular copper sufficiency and exhibit antimicrobial activity." Proceedings of the National Academy of Sciences 118.8 (2021): e2015224118.

Since you first reported iron acquisition systems in the Cladophialophora genus by identifying multiple putative siderophore transporters and the genes related to both siderophore production and RIA systems that are significantly regulated by iron starvation. Analysis of cell extracts also revealed ferricrocin as an intracellular siderophore. Considering the abundance of information presented in this manuscript, it would be valuable to construct a comprehensive model that encompasses all our discoveries, which can be inserted into your discussion. This kind of model will greatly aid readers in comprehending the intricacies of iron acquisition systems within the Cladophialophora genus. I highly recommend it.

The paper is well-written and provides sufficient supportive information, making it easy to follow. 

Author Response

Line 85-86: Please check your citation to make sure they are consistent throughout the manuscript.

Response: Thanks for the comment. The citation format was verified and corrected.

Line 222: Could you please provide another figure with high quality?

Response: Thanks for the observation. A higher quality image was supplied.

Line 326: Can Cladophialophora produce spores? Could you apply spore suspension to do the CAS assay? Please refer to the paper published in PNAS. They use CAS assay for the detection of secreted copper-binding molecules. They ensure the spore concentrations are the same at the beginning. Raffa, Nicholas, et al. "Dual-purpose isocyanides produced by Aspergillus fumigatus contribute to cellular copper sufficiency and exhibit antimicrobial activity." Proceedings of the National Academy of Sciences 118.8 (2021): e2015224118.

Response: Yes, C. carrionii produces spores. In Fig. 5A we present a qualitative assay to show that siderophores are produced and secreted under iron deprived conditions. Although plates of 10 and 15 days were shown, we do not intend to perform a quantitative analysis. Anyway, to avoid miss interpretation, only the 15 days old plate was maintained in the figure and the text was reformulated in accordance.

Since you first reported iron acquisition systems in the Cladophialophora genus by identifying multiple putative siderophore transporters and the genes related to both siderophore production and RIA systems that are significantly regulated by iron starvation. Analysis of cell extracts also revealed ferricrocin as an intracellular siderophore. Considering the abundance of information presented in this manuscript, it would be valuable to construct a comprehensive model that encompasses all our discoveries, which can be inserted into your discussion. This kind of model will greatly aid readers in comprehending the intricacies of iron acquisition systems within the Cladophialophora genus. I highly recommend it.

Response: We thank the reviewer for this valuable insight. A model is now presented as Figure 7 in the revised version of the manuscript.